# In-person training on COVID-19 case management and infection prevention and control: Evaluation of healthcare professionals in Bangladesh

Lubaba Shahrin[1,2]*, Irin Parvin[2], Monira Sarmin[1], Nayem Akhter Abbassi[3], Mst. Mahmuda Ackhter[2], Tahmina Alam[2], Gazi Md. Salahuddin Mamun[2], Aninda Rahman[4], Shamsun Nahar Shaima[2], Shamima Sharmin Shikha[2], Didarul Haque Jeorge[2], Mst. Arifun Nahar[2], Sharifuzzaman[1], Haimanti Saha[1], Abu Sayem Mirza Md Hasibur Rahman[2], Abu Sadat Mohammad Sayeem Bin Shahid[2], A. S. G. Faruque[2], Tahmeed Ahmed[2,5], Mohammod Jobayer Chisti[1,2]

1 Dhaka Hospital, International Centre for Diarrheal Disease Research, Bangladesh (icddr,b), Dhaka, Bangladesh, 2 Nutrition and Clinical Services Division, International Centre for Diarrheal Disease Research, Bangladesh (icddr,b), Dhaka, Bangladesh, 3 National Institute of Mental Health, Dhaka, Bangladesh, 4 Communicable Disease Control (CDC), MoHFW, Dhaka, Bangladesh, 5 Office of the Executive Director, International Centre for Diarrhoeal Disease Research, Bangladesh (icddr,b), Dhaka, Bangladesh

* lubabashahrin@icddrb.org

**Data Availability Statement:** The data set contained personal information of the study participants. Our institutional review board will not

## Abstract

### Background

As COVID-19 was declared a global pandemic, the major focus of healthcare organizations shifted towards preparing healthcare systems to handle the inevitable COVID-19 burden at different phases and levels. A series of in-person training programs were operated in collaboration with government and partner organizations for the healthcare workers (HCW) of Bangladesh. This study aimed to assess the knowledge of HCWs regarding SARS-CoV-2 infection, their case management, infection prevention and control to fight against the ongoing pandemic.

### Methods

As a part of the National Preparedness and Response Plan for COVID-19 in Bangladesh, the training program was conducted at four district-level hospitals and one specialized hospital in Bangladesh from July 1, 2020 to June 30, 2021. A total of 755 HCWs participated in the training sessions. Among them, 357 (47%) were enrolled for the evaluation upon completion of the data, collected from one district hospital (Feni) and one specialized hospital (National Institute of Mental Health).

### Results

The mean percentage of pre-test and post-test scores of all the participants were found to be 57% (95% CI 8.34–8.91; p 0.01) and 65% (95% CI 9.56–10.15; p <0.001) respectively. The difference of score (mean) between the groups was significant (p<0.001). After

have the provision to disclose any kind of information. Thus, our policy is not to make available the data set in the manuscript, the supplemental files, or a public repository. However, data related to this manuscript are available upon request and researchers who meet the criteria for access to confidential data may contact Ms. Armana Ahmed (aahmed@icddrb.org) at the research administration of icddr,b (http://www.icddrb.org).

**Funding:** icddr,b has organized this project with the collaboration of the Government People's Republic of Bangladesh and the development partners. This entire project has been funded by the Global Affairs of Canada (Gr-01686 awarded to LS).

**Competing interests:** No authors have competing interests

categorizing participants' knowledge levels as poor, average and fair, doctors' group has shown to have significant enhancement from level of average to fair compared to that of the nurses. Factors associated with knowledge augmentation of doctors were working in primary health care centers (aOR: 4.22; 95% CI: 1.80, 9.88), job experience less than 5 years (aOR: 4.10; 95% CI: 1.01, 16.63) and experience in caring of family member with COVID-19 morbidity (aOR: 2.06; 95% CI: 1.03, 4.10), after adjusting for relevant covariates such as age, sex and prior COVID-19 illness.

## Conclusion

Considering the series of waves of COVID-19 pandemic with newer variants, the present paper underscores the importance of implementing the structured in-person training program on case management, infection prevention and control for the HCWs that may help for successful readiness prior to future pandemics that may further help to minimize the pandemic related fatal consequences.

## Background

The novel Coronavirus emerged in Wuhan, China in late December 2019, and Bangladesh had reported its first case on 8th March 2020 [1, 2]. Researchers identified the basic reproductive number of the virus, which is higher than many deadly infectious diseases, that eventually overwhelmed healthcare facilities in developing as well as developed nations [3]. The health care system in Bangladesh is typical of any developing country setting and was not fully capable of handling the raging Coronavirus pandemic [4]. However, as the first responders, healthcare workers (HCWs) immediately responded with team efforts in delivering health care and essential services to save lives [5]. In the battle against SARS-CoV-2, Bangladesh has lost nearly 9,400 healthcare workers including 3,106 doctors, 2,281 nurses and 4,015 other individuals related to health service delivery [6]. The sudden spread of the pandemic with fear of contracting the deadly disease abruptly halted all types of in-person skilled training when it was a dire need [2].

In a developing country like Bangladesh, the healthcare workforce is already burdened with routine health care as well as with public health threat issues, including undernutrition, noncommunicable diseases, etc., which might impede adopting preventive measures and facilitating care of COVID-19 [7]. A recent study estimated the high workload of Bangladeshi physicians and nurses with a coverage of 3.04 and 1.07 per 10,000 individuals respectively [8]. Due to heavy workload, HCWs often could not practice and follow the critical preventive health measures such as frequent hand hygiene [9]. Moreover, the case-fatality-rate from COVID-19 among the doctors is highest in Bangladesh compared to other affetcted countries [10]. Considering the intensifying health risk, the Government of Bangladesh conducted a National Preparedness and Response Plan for COVID-19, jointly with international development partners, which indicated the need for hands-on training for doctors, nurses and clinical care support staff [11]. This ranges from prevention of the infection that the health system staff members will be allowed to practice as preventative measures to be consistently followed later by the HCWs to protect themselves and facilitate infection control, and treatment of COVID-19 patients [12].

Despite developing various guidelines for the infection prevention and control procedures and case management of COVID-19 by the Government of Bangladesh, the pandemic

triggered the shut-down of all in-person educational activities. Consequently this placed the unprepared health workforce at utmost risk and simultaneously made them susceptible to spread the transmission [13]. During the COVID-19 pandemic, scientific sessions, conferences and training events were switched to a virtual platform. Although considering a better forum for networking, information exchange and group-based peer learning, we preffered conventional in-person trainings.Under these circumstances, an in-person training program was conducted emphasizing on epidemiology, microbiology, prevention and control and treatment of COVID-19 for the HCWs of different districts of Bangladesh from 1st July 2020 to 30th June 2021 with support from Global Affairs Canada (GAC). Such endeavor was carried out by a dedicated team comprising of physicians, nurses and infection prevention and control specialist from icddr,b.

Literature review indicated that in-person training on an acute emergency disorder encouraged team building in healthcare delivery, leadership and thus improved the performance of caregivers [14]. The objective of our study was to explore the effectiveness of the training program by comparing the pre-test and post-test scores of the participants and formulate better strategies for designing more in-person training in future during any large-scale outbreak in Bangladesh.

## Method

### Study site

The training was conducted in four district and one specialized hospitals (including 21 health facilities). The district hospitals have secondary level health facility (location in Chandpur, Narail, Magura, and Feni) whereas the specialized hospital named National Institute of Mental Health (NIMH) has tertiary level health facility. As per the National Health Service (NHS), medical care is provided in two main ways: primary care (by general practioners and community services) and secondary care (hospitals and specialists). Difference between primary and secondary healthcare: day-to-day healthcare in every local area is considered as primary health care, whereas secondary healthcare is the specialist treatment and support provided by doctors and other health professionals for patients who are referred to them for specific expert care, most often provided in hospitals.

These facilities were selected as they were directly involved in providing optimal care for COVID-19 cases or prepared to hospitalize and manage in case of exceeding the number of cases of COVID-19. Both the health facilities are from urban and sub-urban locality.

### Study participants

Participants were selected by the hospital authority considering their engagement in clinical work and overseeing roles in the team. Moreover, from each hospital, one focal person was selected as an Infection Prevention and Control (IPC) lead who would continue conducting the similar in-house sessions for other staff members after the completion of their training. Among the total 755 participants, 294 were physicians, 335 were nurses and 126 were clinical support staff. If we divide them district wise, the distribution seems like- in Chandpur (n = 79), Narail (n = 141), Magura (n = 97), Feni (n = 264) and, (n = 174) in Dhaka.

We have purposively selected 357 members of the health workforce for the current study, where we aimed to explore the improvement of knowledge by the training program by evaluating their performance in pre-test and post-test assessment sessions.

### Inclusion criteria

Participants from NIMH (specialized hospital) and one district hospital (Feni) were selected for the comparison of this pre and post study at the targeted level of scores.

### Exclusion criteria

We excluded participants from the analysis if either of the pre-test or the post-test performance was incomplete by the respective member of the health workforce.

### Study design

This was a cross-sectional study. The study participants of this analysis were the individuals who received training in a COVID-19 training program that aimed at the capacity development of healthcare workers in taking care of COVID-19 cases in Bangladesh.

### Instruments

A self-directed, semi-structured questionnaire (S1 Questionnaire) was developed by the investigators of the study where identity of the participants was masked. The questionnaire was in English and attached as a S1 Questionnaire. We have used the identical 15 questions for both pre-test and post-test. Thus four domains (epidemiology, clinical manifestation, case management, and infection prevention and control) had 3, 2, 2 and 8 questions respectively. We categorized the knowledge as poor, average and fair if the participant scored 0–5, 6–10 and 11–15 respectively.The questionnaire was validated in-house among trainee physicians of Dhaka Hospital of icddr,b as a piloting approach. In addition, there were questions about the respondent's age, sex, educational background, workplace, current working position with joining date, COVID-19 morbidity experience (either self-infection or the infection of a family member) and their feedback regarding their liking and disliking of the training. All the course materials were provided to the participants in the form of handout of presentation.

### Description of training

As a part of the National Preparedness and Response Plan for COVID-19 in Bangladesh, a rapid facility assessment for national health facility readiness and preparedness for COVID-19 was commissioned. The Directorate General of Health Services (DGHS) initiated the process with support from development partners, UN agencies, and other national and international agencies. USAID's Medicine, Technologies and Pharmaceuticals Services (MTaPS) Program, implemented in Bangladesh by Management Sciences for Health (MSH), supported DGHS with the assessment. It was a two-day session for doctors, one day session for nurses and other support staff members.

For doctors and nurses, different sets of training curriculum and training materials were used as appropriate considering the compatible topics and modules focusing on their levels of understanding. After each module, a comforting refreshment break was arranged in every training session. A flexible approach was adopted for delivering the training materials, that included theoretical sessions, practical demonstrations and interactive discussions.

### Curriculum of the training

The goal of the training was to empower the healthcare workforce to fight COVID-19 by achieving confident, skills and knowledge and simultaneously implementing evidence-based strategies in healthcare settings. Thus, the training modules were designed to document knowledge, attitude and practice of the participants related to COVID-19 pandemic.

The IPC modules included the importance of triage, principles of IPC, standard precautions and transmission-based precautions, waste management and environmental decontamination and vaccine development. The case management modules included: the epidemiology of SARS-CoV-2; clinical manifestations of COVID-19; diagnostic strategy and treatment of COVID-19 cases.

## Sample size

We have collected information from 357 participants (214 from district level hospitals and 143 from a tertiary level/specialized hospital) after excluding 12 participants (8 and 4, respectively) because of their incomplete response in either of the two sessions.

Assuming 20% of the participants might lack in improvement of knowledge and skill after the training sessions, the estimated sample size with 80% power at a 95% confidence limit and 5% precision was 246, which is convenient for the present evaluation. The evaluation was purposively selected after meeting the exclusion criteria based on unintended to comply with providing personal and job related information

## Statistical analysis

The data entry, coding, and editing were performed using SPSS version 20 (Chicago, IL, USA). The SPSS file was then imported into Stata 15 software and all statistical analyses were performed using Stata (Stata Statistical Software: Release 15, College Station, Texas 77,845, USA: Stata Corp LLC). Descriptive statistics included frequencies, percentages, mean, standard deviation which were used to summarize data. Statistical plots such as cluster diagram, and pie charts were reviewed for data visualization. To estimate the inferential statistics, odds ratio with 95% confidence interval was used. We compared the baseline characteristics of the training participants by using chi-square tests, and t-test was used to see the significance of mean difference between pre-test and post-test scores. Logistic regression analysis revealed the more distinctive association between participants' characteristics and pre-test and post-test scores.

## Ethical approval

Ethical approval for the training activity was obtained from the Research Review Committee and Ethical Review Committee of the International Centre for Diarrhoeal Diseases Research, Bangladesh (icddr,b)(ACT-01112) on 25th June 2020. All participants provided voluntary written consent and actively participated in the pre-test and post-test sessions.

## Results

From 1st July 2020 to 30th June 2021, a total of 755 health care providers participated in the training sessions. Among them, 357 (47%) participants (doctors and nurses) were assessed by the pre-test and post-test. Out of the 357, 208 (58.26%) were nurses with a diploma in Nursing (72%). Among the physicians, nearly one-fourth had specialization either in medicine or surgery (Fig 1).

Table 1 summarizes the characteristics of the study population. The mean age of the population was (32.82±4.99) years for doctors and (35.46±8.31) years for nurses. About 72% of the total respondents were female. More than half (59.94%) of the participants were from primary and secondary level healthcare facilities. Approximately 80% of doctors had an average working experience of less than five years, whereas one-third of the nurses had a working experience of more than ten years (Table 1). The doctors suffered from COVID-19 more often than the nurses as well as they had witnessed sufferings of family members more frequently than the nurses.

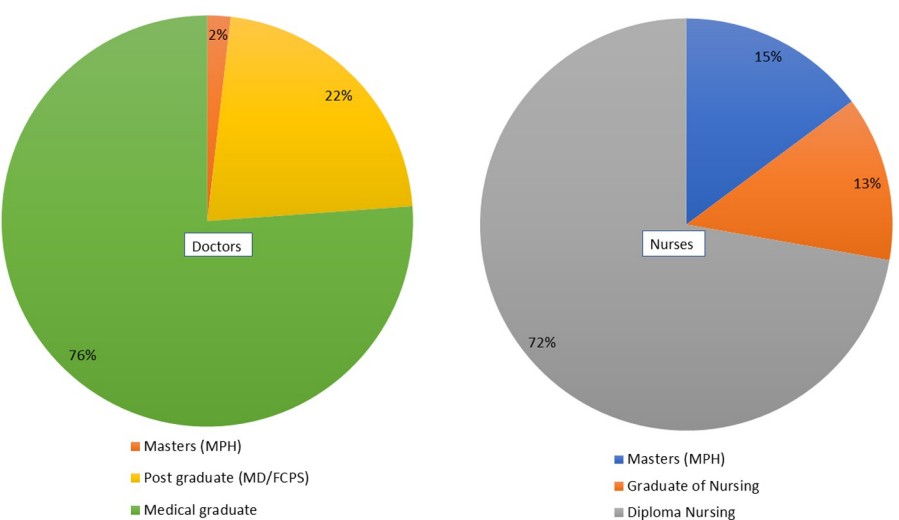

**Fig 1. Academic contexts of the participating doctors and nurses.**

Table 2 shows the participants' overall pre-test and post-test scores according to their current working location. Participants working in the primary healthcare facility achieved higher scores in pre-test and post-test assessments than the other participants from secondary or tertiary level facilities. Nurses working in the primary healthcare facilities achieved higher post-test scores than the other nurses. However, doctors working in all three settings achieved similar post-test scores of more than 80%.

The knowledge levels regarding epidemiology, clinical manifestation, case management and infection prevention and control are presented in Table 3. The results observed that all the participants achieved more than 50% pre-test scores and significantly higher post-test scores in all four domains.

We have presented the knowledge of the participants into three categories. Table 4 shows the distribution of participants' knowledge level according to the characteristics of the participants before and after conducting the training. There was an overall significant improvement

**Table 1. Demographic and occupational characteristics of the participants.**

| Variables | Overall (N = 357) | Doctors (n = 149) (%) | Nurses (n = 208) (%) | *p*-value |
|---|---|---|---|---|
| Age, mean±SD | - | 32.82±4.99 | 35.46±8.31 | 0.005 |
| Female sex | 256 | 64 (42.95) | 192 (92.31) | <0.001 |
| Training site, n (%) | 357 | | | |
| Specialized hospital | 143 | 53 (35.57) | 90 (43.27) | 0.143 |
| District hospital | 214 | 96 (64.43) | 118 (56.73) | |
| Current working station, n (%) | 357 | | | |
| Primary health care facility (sub-district facility) | 118 | 61 (40.94) | 57 (27.40) | 0.027 |
| Secondary health care facility (district facility) | 96 | 35 (23.49) | 61 (29.33) | |
| Years of experience, n (%) | 217 | | | |
| <5 years | | 91 (81.25) | 59 (56.19) | <0.001 |
| 5–10 years | | 16 (14.29) | 12 (11.43) | |
| >10 years | | 5 (4.46) | 34 (32.38) | |
| COVID-19 infection, n (%) | 67 | 42 (37.84) | 25 (23.81) | 0.026 |
| COVID-19 exposure from the family, n (%) | 55 | 50 (45.05) | 5 (22.73) | 0.052 |

**Table 2. Current working station wise comparison of pre-test and post-test scores of the participants.**

| | | Pre-test score (n = 357), Mean % | Post-test score (n = 357), Mean % | p value |
|---|---|---|---|---|
| **Primary Healthcare Centre** | Overall | 63.39 | 71.58 | <0.001 |
| | Doctors | 75.30 | 81.64 | <0.001 |
| | Nurses | 50.64 | 60.82 | <0.001 |
| **Secondary Healthcare Centre** | Overall | 56.18 | 65.21 | <0.001 |
| | Doctors | 72.95 | 81.33 | <0.001 |
| | Nurses | 46.56 | 55.96 | <0.001 |
| **Tertiary Healthcare Centre** | Overall | 53.61 | 61.17 | <0.001 |
| | Doctors | 68.05 | 80.38 | <0.001 |
| | Nurses | 45.11 | 49.85 | <0.001 |

in the scores from an average level to a fair level. A significant improvement of scores from an average level to acceptable fair level was observed among participants of all categories except those who experienced COVID-19 morbidity.

Fig 2 shows the categorical evaluation of the scores of both doctors and nurses. There is an obvious improvement of knowledge score for the doctors compared to the nurses. There was a significant decrease in the proportion of average level in post-test (11%) than pre-test (40%) ($p<0.001$) and increased significantly in the proportion of fair level in post-test (87%) than pre-test (59%). On the other hand, there was no remarkable change in the scores of the nurses.

Results of unadjusted (uOR) and adjusted odds ratios (aOR) computed by logistic regression are shown in Table 5. The observed improvement is reflected as an increase in scores in case of doctors (aOR: 16.27; 95% CI: 5.68, 46.56). Participants currently working in a secondary healthcare facility (aOR: 2.88; 95% CI: 1.07, 7.72) and primary healthcare facility (aOR: 4.22; 95% CI: 1.80, 9.88), having working experience of 5–10 years (aOR: 4.72; 95% CI: 1.15, 19.39) or less than 5 years (aOR: 4.10; 95% CI: 1.01, 16.63) were more likely to achieve a better post-test score. Participants who experienced in caring of a family member with COVID-19 (aOR: 2.06; 95% CI: 1.03, 4.10) achieved significantly higher post-test scores (Table 5).

## Discussion

The COVID-19 pandemic has revealed deep inequities in the global healthsystem to mitigate health emergency prevention response. A number of independent reviews have identified numerous gaps and weaknesses in the health system preparedness for any pandemic, moreover the condition is worse in developing country.The present study underpins one of the very few published studies assessing knowledge and preventive behaviors of healthcare workforce towards COVID-19 management in Bangladesh. Government reports have identified high COVID-19 burden districts of Bangladesh, where in-person training impacted meaningfully in managing COVID-19 cases [15]. The major strength of our research paper is the larger sample size and district-wise distribution of working-location of participants, which provided a

**Table 3. Comparison of pre-test and post-test scores of all the participants on the basis of four domains of different questions.**

| Domain Name | Pre-test score (n = 357), Mean % | Post-test score (n = 357), Mean % | p value |
|---|---|---|---|
| **Epidemiology** | 72.36 | 82.17 | <0.001 |
| **Clinical manifestations** | 52.24 | 62.04 | <0.001 |
| **Case management** | 56.44 | 67.65 | <0.001 |
| **Infection prevention and control** | 53.61 | 59.98 | <0.001 |

**Table 4. Comparative characteristics of the participants between different results level of pre-test and post-test.**

| | Results | Pre-test (n = 357) | Post-test (n = 357) | *p*-value |
|---|---|---|---|---|
| | | Frequency (%) | Frequency (%) | |
| **Overall** | **Poor level (0–5)** | 41(11.48) | 27(7.56) | *0.074* |
| | **Average level (6–10)** | 210(58.82) | 173(48.46) | ***0.005*** |
| | **Fair level (11–15)** | 106(29.69) | 157(43.98) | ***0.000*** |
| **Age (<35)** | **Poor (0–5)** | 11(7.80) | 7(4.96) | *0.330* |
| | **Average (6–10)** | 70(49.65) | 58(41.13) | *0.151* |
| | **Fair (11–15)** | 60(42.55) | 76(53.90) | *0.057* |
| **Age (≥35)** | **Poor (0–5)** | 16(21.62) | 9(12.16) | *0.125* |
| | **Average (6–10)** | 43(58.11) | 36(48.65) | *0.249* |
| | **Fair (11–15)** | 15(20.27) | 29(39.19) | ***0.012*** |
| **Male** | **Poor (0–5)** | 3(2.97) | 3(2.97) | *1.000* |
| | **Average (6–10)** | 43(42.57) | 22(21.78) | ***0.002*** |
| | **Fair (11–15)** | 55(54.46) | 76(75.25) | ***0.002*** |
| **Female** | **Poor (0–5)** | 38(14.84) | 24(9.38) | *0.058* |
| | **Average (6–10)** | 167(65.23) | 151(58.98) | *0.145* |
| | **Fair (11–15)** | 51(19.92) | 81(31.64) | ***0.002*** |
| **Tertiary Healthcare Center** | **Poor (0–5)** | 22 (15.38) | 15(10.49) | *0.217* |
| | **Average (6–10)** | 91(63.64) | 81(56.64) | *0.227* |
| | **Fair (11–15)** | 30(20.98) | 47(32.87) | ***0.023*** |
| **Primary and Secondary Healthcare Center** | **Poor (0–5)** | 19(8.88) | 12(5.61) | *0.192* |
| | **Average (6–10)** | 119(55.61) | 92(42.99) | ***0.009*** |
| | **Fair (11–15)** | 76(35.51) | 110(51.40) | ***0.001*** |
| **COVID-19 infection** | **Poor (0–5)** | 5(7.46) | 4(5.97) | *0.730* |
| | **Average (6–10)** | 33(49.25) | 25(37.31) | *0.163* |
| | **Fair (11–15)** | 29(43.28) | 38(56.72) | *0.120* |
| **Not-infected with COVID-19** | **Poor (0–5)** | 22(14.77) | 13(8.72) | *0.105* |
| | **Average (6–10)** | 81(54.36) | 69(46.31) | *0.164* |
| | **Fair (11–15)** | 46(30.87) | 67(44.97) | ***0.012*** |
| **COVID-19 exposure from the family** | **Poor (0–5)** | 1(1.82) | 2(3.64) | *0.558* |
| | **Average (6–10)** | 21(38.18) | 6(10.91) | ***0.001*** |
| | **Fair (11–15)** | 33(60.00) | 47(85.45) | ***0.003*** |
| **Not exposed to COVID-19 from family** | **Poor (0–5)** | 4(5.13) | 1(1.28) | *0.173* |
| | **Average (6–10)** | 37(47.44) | 23(29.49) | ***0.021*** |
| | **Fair (11–15)** | 37(47.44) | 54(69.23) | ***0.006*** |

better insight of the scenario towards COVID-19 care in Bangladesh. The aim of our study was to quantify the knowledge gained through the COVID-19 on-site training in Bangladesh.

Our study observed female-dominated cluster of participants. However, both males and females performed equally in assessment tests. While searching for the gender-wise acquaintances of knowledge on COVID-19, reports showed that male participants achieved better knowledge than their female counterpart [16, 17], which differed in our research.

Due to easier accessibility of information in urban areas, previous studies have reported a higher level of knowledge among individuals living in urban areas [16, 18], although our study has shown contrasting findings. Participants assigned to rural healthcare facilities like primary healthcare facilities (sub-district health facilities) and, secondary health care facilities (district health facilities) have demonstrated better performance in training assessment. This can be

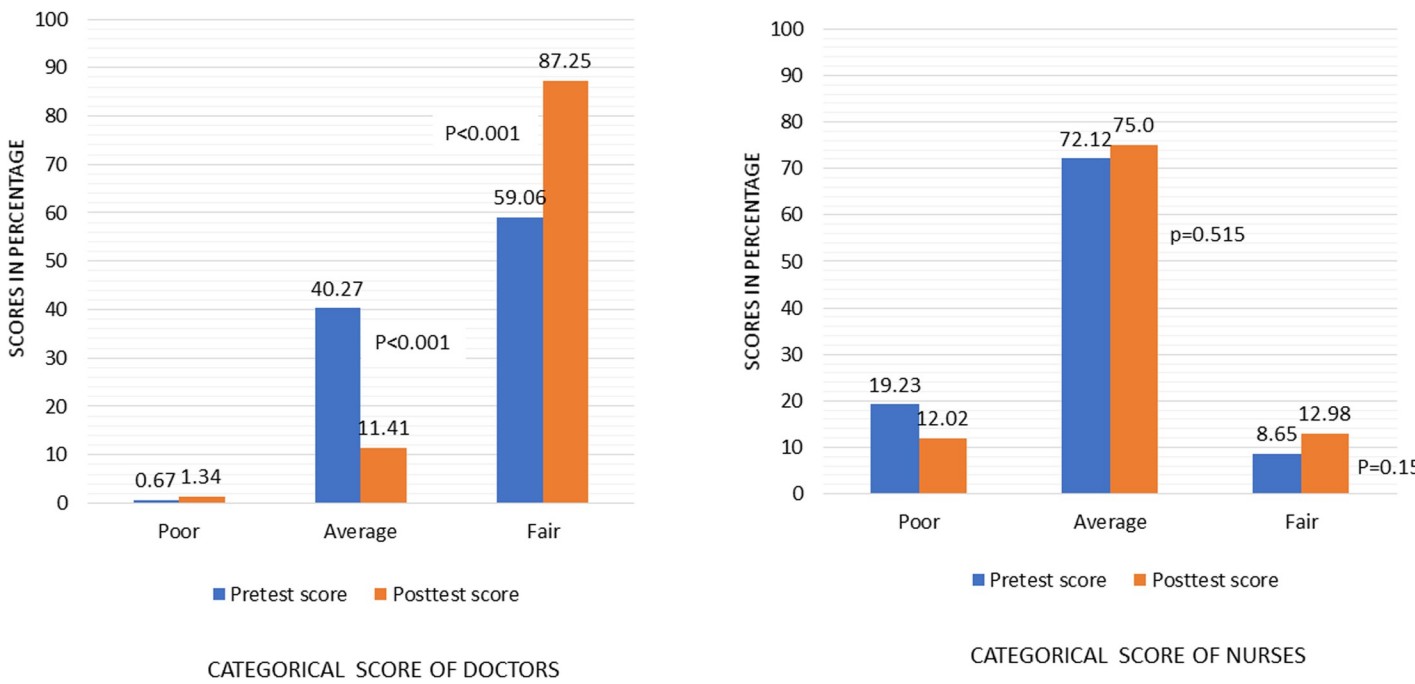

**Fig 2. Pretest-posttest score level of doctors and nurses.**

explained by the knowledge saturation of the urban participants through easier accessibility of educational modules on COVID-19 [19].

Our study explored the association of length of working experience with better knowledge gain among the participants. The study revealed the fact that knowledge gain was inversely related to the year of job experience. From the Bangladesh Government civil service perspective, it is prudent to mention that those who have fewer years of job experience belong to younger age group. It is prudent to mention that young people are wel-motivated in acquiring and systhesising of new knowledge than the older people. Moreover, there were some restriction of in-person involvement of patient care in the age group of over forty. As younger health care workers involved actively in patient care from the beginning of pandemic, they were more engaged in the training, hence acquired better knowledge compared to their older peers. In other word, it is obvious from previous literatures that young people are better achievers of learning new knowledge and skills [20] compared to the aged people [21]. Moreover, participants exposed to a COVID-19 sick family member performed better in the assessment tests. Previous studies also reported that healthcare workers were in constant mental worries for potential transmission of the virus to their family members [22] and therefore, would be better experienced in COVID-19 patient care if having practical experience in caring for their sick family members [23].

Nurses and clinicians are two major pillars of any healthcare facility and are instrumental in the case management of COVID-19 pandemic. In particular, it is necessary to assess the knowledge gained in a conventional way of in-person training, which is followed by an impact assessment of the training, to ascertain the future needs of the training program during a pandemic situation. The study's outcome could be helpful while developing an effective and productive infrastructure.

The study is not devoid of limitations. Firstly, we could not include all the participants who received training that could have resulted in difference in the evaluation. In addition, we

**Table 5. Associated factors of the participants with pretest and posttest scores.**

| Indicators | Unadjusted OR (95% CI) | p-value | Adjusted OR (95% CI) | p-value |
|---|---|---|---|---|
| **Age** | | | | |
| ≥35 | Reference | | Reference | |
| <35 | 2.20 (1.44, 3.36) | <0.001 | 1.16 (0.52, 2.59) | 0.711 |
| **Sex** | | | | |
| Female | Reference | | Reference | |
| Male | 5.31 (3.74, 7.54) | <0.001 | 1.38 (0.70, 2.72) | 0.348 |
| **Position** | | | | |
| Nurses | Reference | | Reference | |
| Doctors | 22.47 (15.03, 33.57) | <0.001 | 16.27 (5.68, 46.56) | <0.001 |
| **Current working station** | | | | |
| Tertiary Healthcare Center | Reference | | Reference | |
| Secondary Healthcare Center | 1.49 (1.00, 2.21) | 0.048 | 2.88 (1.07, 7.72) | 0.036 |
| Primary Healthcare Center | 2.71 (1.88, 3.91) | <0.001 | 4.22 (1.80, 9.88) | 0.001 |
| **Years of experience** | | | | |
| >10 years | Reference | | Reference | |
| 5–10 years | 10.14 (3.98, 25.88) | <0.001 | 4.72 (1.15, 19.39) | 0.032 |
| <5 years | 9.87 (4.40, 22.17) | <0.001 | 4.10 (1.01, 16.63) | 0.049 |
| **COVID-19 infection** | | | | |
| No | Reference | | Reference | |
| Yes | 1.64 (1.08, 2.47) | 0.019 | 1.07 (0.56, 2.07) | 0.831 |
| **COVID-19 exposure from the family** | | | | |
| No | Reference | | Reference | |
| Yes | 1.90 (1.12, 3.23) | 0.016 | 2.06 (1.03, 4.10) | 0.041 |

After adjusting for relevant covariates, there was no significant association between scores and the participants by age, sex, and who infected by COVID-19.

assessed the effectiveness of the on-site training program by performing pre-test and post-test that can measure only short-term memory. On the other hand, the real-time practice of IPC and COVID-19 management was not evaluated through observations for a sufficient time period in case of this training. Lastly, we were not able to adjust for the changes in trainers for different modules of the training.

In conclusion, the results of our training clearly indicate that the HCWs were benifitted by attending structured training on COVID-19 case management, infection prevention and control. To nurture their expertise more training program need to be developed and implemented on both preventive and curative care of COVID-19 patients in different hospitals that may further help for rapid preparedness and optimal implementation planning for any pandemic similar to COVID-19.

## Supporting information

**S1 Questionnaire.**
(PDF)

**S1 Table. Change of knowledge between pre-test and post-test among participants.**
(DOCX)

## Author Contributions

**Conceptualization:** Lubaba Shahrin, Aninda Rahman, Abu Sadat Mohammad Sayeem Bin Shahid, A. S. G. Faruque, Tahmeed Ahmed.

**Data curation:** Lubaba Shahrin, Monira Sarmin, Nayem Akhter Abbassi, Tahmina Alam, Shamsun Nahar Shaima, Shamima Sharmin Shikha, Didarul Haque Jeorge, Mst. Arifun Nahar.

**Formal analysis:** Lubaba Shahrin, Monira Sarmin, Tahmina Alam, Gazi Md. Salahuddin Mamun, Mst. Arifun Nahar, Mohammod Jobayer Chisti.

**Funding acquisition:** Lubaba Shahrin, Tahmeed Ahmed.

**Investigation:** Lubaba Shahrin, Nayem Akhter Abbassi, Tahmina Alam, Gazi Md. Salahuddin Mamun, Aninda Rahman, Haimanti Saha.

**Methodology:** Lubaba Shahrin, Monira Sarmin, Shamsun Nahar Shaima, Mst. Arifun Nahar, Sharifuzzaman, A. S. G. Faruque.

**Project administration:** Lubaba Shahrin, Mst. Mahmuda Ackhter, Aninda Rahman, Didarul Haque Jeorge, Abu Sayem Mirza Md Hasibur Rahman, Abu Sadat Mohammad Sayeem Bin Shahid.

**Resources:** Lubaba Shahrin, Irin Parvin, Nayem Akhter Abbassi, Gazi Md. Salahuddin Mamun.

**Software:** Irin Parvin, Gazi Md. Salahuddin Mamun, Mst. Arifun Nahar.

**Supervision:** Lubaba Shahrin, Mst. Mahmuda Ackhter, Aninda Rahman, Shamsun Nahar Shaima, Abu Sayem Mirza Md Hasibur Rahman, Abu Sadat Mohammad Sayeem Bin Shahid, A. S. G. Faruque, Tahmeed Ahmed, Mohammod Jobayer Chisti.

**Validation:** Irin Parvin, Nayem Akhter Abbassi, Mst. Arifun Nahar, Sharifuzzaman, A. S. G. Faruque, Mohammod Jobayer Chisti.

**Visualization:** Mst. Mahmuda Ackhter, Haimanti Saha.

**Writing – original draft:** Lubaba Shahrin, Irin Parvin, Monira Sarmin, Nayem Akhter Abbassi, Mst. Mahmuda Ackhter, Tahmina Alam, Gazi Md. Salahuddin Mamun, Aninda Rahman, Shamsun Nahar Shaima, Shamima Sharmin Shikha, Didarul Haque Jeorge, Sharifuzzaman, Haimanti Saha, Abu Sayem Mirza Md Hasibur Rahman, Abu Sadat Mohammad Sayeem Bin Shahid, A. S. G. Faruque, Tahmeed Ahmed, Mohammod Jobayer Chisti.

**Writing – review & editing:** Lubaba Shahrin, Mohammod Jobayer Chisti.

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
