## [Decision Letter · Decision Letter 0]

6 Apr 2022

PONE-D-22-01969Evaluation of structured in-person COVID-19 case management and infection prevention and control training for healthcare professionals in BangladeshPLOS ONE

Dear Dr. Shahrin,

Thank you for submitting your manuscript to PLOS ONE. After careful consideration, we feel that it has merit but does not fully meet PLOS ONE’s publication criteria as it currently stands. Therefore, we invite you to submit a revised version of the manuscript that addresses the points raised during the review process.

Your manuscript has been reviewed by two experts in the field. Both have expressed concerns regarding the overall quality of manuscript including editing concerns. I do agree with the issues that they have raised. I urge you to carefully address the comments and submit the manuscript for another round of review. 

We look forward to receiving your revised manuscript.

Kind regards,

Sanjai Kumar

Academic Editor

PLOS ONE

Journal Requirements:

“icddr,b has organized this project with the collaboration of the Government People’s Republic of Bangladesh and the development partners. This entire project has been funded by the Global Affairs of Canada (GR-01686).”

Reviewers' comments:

Reviewer's Responses to Questions

**Comments to the Author**

1. Is the manuscript technically sound, and do the data support the conclusions?

Reviewer #1: Yes

Reviewer #2: Yes

2. Has the statistical analysis been performed appropriately and rigorously? 

Reviewer #1: Yes

Reviewer #2: Yes

3. Have the authors made all data underlying the findings in their manuscript fully available?

Reviewer #1: No

Reviewer #2: Yes

4. Is the manuscript presented in an intelligible fashion and written in standard English?

Reviewer #1: No

Reviewer #2: Yes

5. Review Comments to the Author

Reviewer #1: The manuscript “Evaluation of structured in-person COVID-19 case management and infection prevention and control training for healthcare professionals in Bangladesh” by Shahrin et. al., describes the importance of implementing the structured in-person training program for healthcare workers regarding case management in a pandemic.

Authors showed all participants achieved significantly higher post-test scores in Epidemiology, Clinical manifestation, case management and infection prevention and control.

The manuscript needs editorial assistance because of some grammatical mistakes.

Comments are listed below:

1. Authors mentioned semi-structured questionnaire developed by the investigators are in the supplementary file. But this reviewer couldn’t find any.

2. Authors preferred conventional in-person trainings. Is there any virtual training data?

3. Please mention if these healthcare facilities are from rural/sub-urban/urban area.

4. One third of the participants remained in poor post-test score category. About 6% and 9% participants demoted from average and fair category respectively. Did the authors supply any quick access guide?

5. Figures 1 & 2 are of poor quality. This reviewer couldn’t read the scale or what is written there. Please use high quality figure.

6. Do not mention COVID-19 as COVID-10.

7. Please check the typographical errors throughout.

8. Shortened the Title.

Reviewer #2: SUMMARY

The aim of this study was to measure levels of knowledge and preparedness to manage COVID-19 infections among physicians and nurses in Bangladesh.

The COVID-19 pandemic not only puts healthcare workers (HCWs) who treat patients in critical care at high risk but also challenges healthcare systems to respond to the crisis. Adequate training in preventive measures and control are critical for workers’ preparedness and to develop the capacity to handle the inevitable heath care emergency.

In this work, the authors explored the effectiveness of a structured in-person training program directed to HCWs comparing participants’ test scores (pre and post training) with the objective to plan better strategies to effectively navigate current and future large-scale disease outbreaks in Bangladesh.

A similar study was conducted by Muhammed Elhadi et al. in Libya, and was published in 2020 in The American Journal of Tropical Medicine and Hygiene (DOI: https://doi.org/10.4269/ajtmh.20-0330).

The training program was conducted in hospitals from July1, 2020 to June 30, 2021. Seven hundred and fifty-five HCWs participated in the study, which included doctors and nurses. Participants were tested before and after the training, and their knowledge was scored as poor, average and fair.

The results highlight the benefit of implementing structured in-person education and training programs for HCWs, which contributes to successful case management and COVID-19 infection prevention and control. In addition, the authors showed the importance of organized and collaborative partnership between Bangladesh government and international organizations.

MERITS

Critical preparedness, readiness, and knowledge regarding COVID-19 are needed for physicians and nurses working on the front line. Few researchers have addressed the overall issues of preparedness of healthcare systems for COVID-19 and their ability to maintain control of the epidemic. The study highlights the importance of educational initiatives to help countries improve their capacity to control and prevent COVID-19 infection.

COMMENTS

1. The manuscript has language and editing issues in all sections. I suggest the authors work with a scientific editor to improve the flow and readability of the text.

2. Background

a) Lines 128-133: the statements included in this paragraph do not connect well. I suggest the authors delete the first sentence (128-130). The information about the study participants is already included in the methods section.

b) Please define the acronym icddr,b.

3. Methods

c) Sites and study participants:

• Information about participants and sites is mixed up and it’s difficult to follow. Splitting the section in two sub-sections (1. Sites; 2. Participants) maybe helpful for the reader and reviewers to understand the study characteristics.

• Please state the difference between primary and secondary facilities.

• Are the secondary and tertiary level facilities included in the district and specialized hospitals? Please clarify.

d) Study design: in which language was the questionnaire administered? Please include a copy as supplementary material.

e) Training description:

• The information provided in lines 188-189 about the health care facilities is redundant. It was cited in line 186 and in the sites and participants section.

• This sub-section needs editing. The information provided is mixed up (facilities with participants) and therefore it’s difficult to follow.

f) Training curriculum: the information provided in lines 205-207 is redundant because it was already included in precedent sub-sections.

g) Sample size:

• Please explain/edit the sentence starting on line 216: “We could not communicate…”

• Please explain/edit the sentence starting on line 218: “The sample size was adequate…” how did you come to the conclusion that the sample size was adequate?

I suggest the authors move this information to the results section.

4. Results

h) Please note that the quality of Figures 1 and 2 is not good enough for the reader to see the numbers or legends.

i) Line 250: please explain the meaning of the word “destruction” in this sentence.

j) Line 268-270: please clarify this sentence.

k) Line 273: please change “mentioned” to “shown”.

l) Table 1: please change “COVID-19 sufferer” by “COVID-19 infection”

m) Table 4:

• Please change “COVID-19 sufferer” by “COVID-19 infection”

• Please change “not-sufferer by COVID-19 ” by “Non-infected with COVID-19”

n) Table 5:

• Please explain the use of the term “reference” in this table.

• Please change “COVID-19 sufferer” by “COVID-19 infection”.

5. Discussion

o) Line 286-287: please clarify what you want to communicate with “…hospitals will not meet

the entire epidemic burden”.

p) Line 289: please clarify to which study you refer with “the major strength is the ….”.

q) Line 308: did you also find that young people were better achievers than aged people? Please clarify this paragraph.

r) Lines 327-330: this sentence is too long and needs editing.

6. PLOS authors have the option to publish the peer review history of their article (what does this mean?). If published, this will include your full peer review and any attached files.

Reviewer #1: No

Reviewer #2: No

---

## [Author Response · Author response to Decision Letter 0]

20 Apr 2022

Cover letter

Date: April 20, 2022

To: Dr Sanjai Kumar

 Academic Editor, PLoS ONE

From: Dr Lubaba Shahrin

 Corresponding author

 Manuscript Ref. : PONE-D-22-01969

Subject: Response on the comments of the academic editor and the reviewers of PLoS ONE on manuscript Ref.: PONE-D-22-01969 entitled “In-person training in COVID-19 Case management and infection prevention and control: Evaluation of healthcare professionals’ knowledge in Bangladesh”

Dear Dr Sanjai Kumar,

Thank you for providing us the opportunity to resubmit our manuscript following revision. We greatly appreciate the helpful comments of the reviewers and have attempted to address them in full. We now submit two drafts of the revised manuscript, one clean version and one which highlights the changes that we have made in red, as well as this letter which provides a point-by-point response to the reviewers’ comments. 

Kindly consider our revised draft for further in your prestigious journal. Please find the reviewers response as below.

Reviewer’s comments:

Reviewer #1: The manuscript “Evaluation of structured in-person COVID-19 case management and infection prevention and control training for healthcare professionals in Bangladesh” by Shahrin et. al., describes the importance of implementing the structured in-person training program for healthcare workers regarding case management in a pandemic.

Authors showed all participants achieved significantly higher post-test scores in Epidemiology, Clinical manifestation, case management and infection prevention and control.

The manuscript needs editorial assistance because of some grammatical mistakes.

Response: Thank you for your kind concern. We have reviewed the draft by our in-house scientific editor. We hope this draft is now in better shape.

Comments are listed below:

1. Authors mentioned semi-structured questionnaire developed by the investigators are in the supplementary file. But this reviewer couldn’t find any.

Response: We apologies for the inconvenience. The questionnaire is attached here with the supplementary files

2. Authors preferred conventional in-person trainings. Is there any virtual training data?

Response: Although we found some research on knowledge, attitude and practice elsewhere, we do not have any data on virtual or online training. In in-person training we were able to arrange some hands-on activity on infection prevention and practice, which would be missed in virtual training. For your kind convenience we have put one article on online training:

Knowledge, attitudes, and fear of COVID-19 during the Rapid Rise Period in Bangladesh

M. A. Hossain, M. I. K. Jahid, K. M. A. Hossain, L. M. Walton, Z. Uddin, M. O. Haque, et al.

PloS one 2020 Vol. 15 Issue 9 Pages e0239646

3. Please mention if these healthcare facilities are from rural/sub-urban/urban area.

Response: Thank you for your query. In the study site we have clarify about the level of health facilities in Bangladesh. The district level hospital is a secondary level health facility. Both are from urban facilities.

4. One third of the participants remained in poor post-test score category. About 6% and 9% participants demoted from average and fair category respectively. Did the authors supply any quick access guide?

Response: We supplied the all the presentation handout delivered in the training. We have included this in study instrument section.

5. Figures 1 & 2 are of poor quality. This reviewer couldn’t read the scale or what is written there. Please use high quality figure.

Response: We regret for the inconvenience, we have re-created the figures of 300 dpi as per journal’s requirement for your kind consideration.

6. Do not mention COVID-19 as COVID-10.

Response: We apologies for the typing error.

7. Please check the typographical errors throughout.

Response: We have checked and corrected them diligently.

8. Shortened the Title.

Response: Thank you for your suggestions. We have modified the title as “In-person training on COVID-19 case management and infection prevention and control: Evaluation of healthcare professionals in Bangladesh” for your kind contemplation. 

Reviewer #2: SUMMARY

The aim of this study was to measure levels of knowledge and preparedness to manage COVID-19 infections among physicians and nurses in Bangladesh.

The COVID-19 pandemic not only puts healthcare workers (HCWs) who treat patients in critical care at high risk but also challenges healthcare systems to respond to the crisis. Adequate training in preventive measures and control are critical for workers’ preparedness and to develop the capacity to handle the inevitable heath care emergency.

In this work, the authors explored the effectiveness of a structured in-person training program directed to HCWs comparing participants’ test scores (pre and post training) with the objective to plan better strategies to effectively navigate current and future large-scale disease outbreaks in Bangladesh.

A similar study was conducted by Muhammed Elhadi et al. in Libya, and was published in 2020 in The American Journal of Tropical Medicine and Hygiene (DOI: https://doi.org/10.4269/ajtmh.20-0330).

Response: Thank you for kindly sharing this important article that we have also included in the first draft (described in line109 in the introduction with the reference number 7). 

The training program was conducted in hospitals from July1, 2020 to June 30, 2021. Seven hundred and fifty-five HCWs participated in the study, which included doctors and nurses. Participants were tested before and after the training, and their knowledge was scored as poor, average and fair.

The results highlight the benefit of implementing structured in-person education and training programs for HCWs, which contributes to successful case management and COVID-19 infection prevention and control. In addition, the authors showed the importance of organized and collaborative partnership between Bangladesh government and international organizations.

Response: We are very grateful to the reviewer for their treasuring comments.

MERITS

Critical preparedness, readiness, and knowledge regarding COVID-19 are needed for physicians and nurses working on the front line. Few researchers have addressed the overall issues of preparedness of healthcare systems for COVID-19 and their ability to maintain control of the epidemic. The study highlights the importance of educational initiatives to help countries improve their capacity to control and prevent COVID-19 infection.

COMMENTS

1. The manuscript has language and editing issues in all sections. I suggest the authors work with a scientific editor to improve the flow and readability of the text.

Response: Thank you for your suggestions. We have reviewed this draft by our in-house scientific editor, identical native language fluency. 

2. Background

a) Lines 128-133: the statements included in this paragraph do not connect well. I suggest the authors delete the first sentence (128-130). The information about the study participants is already included in the methods section.

Response: Thanks for the suggestion, it is deleted.

b) Please define the acronym icddr,b.

Response: The full form is International Centre for Diarrheal Disease Research, Bangladesh (icddr,b) mentioned in line 29 and line 236.

3. Methods

c) Sites and study participants:

• Information about participants and sites is mixed up and it’s difficult to follow. Splitting the section in two sub-sections (1. Sites; 2. Participants) maybe helpful for the reader and reviewers to understand the study characteristics.

Response: Thanks for the suggestion. We have done accordingly

• Please state the difference between primary and secondary facilities.

Response: Included in the study site from line 141-146

• Are the secondary and tertiary level facilities included in the district and specialized hospitals? Please clarify.

Response: The district hospitals are secondary level health facility and specialized hospitals are tertiary level health facility. Included in the above-mentioned section.

d) Study design: in which language was the questionnaire administered? Please include a copy as supplementary material.

Response: The language is in English. The questionnaire was included in supplementary material

e) Training description:

• The information provided in lines 188-189 about the health care facilities is redundant. It was cited in line 186 and in the sites and participants section.

Response: Thank you for noticing this, we have removed this.

• This sub-section needs editing. The information provided is mixed up (facilities with participants) and therefore it’s difficult to follow.

Response: We apologies, now we have sorted the sections and remove the duplications.

f) Training curriculum: the information provided in lines 205-207 is redundant because it was already included in precedent sub-sections.

Response: Thank you for the suggestion, we have corrected this.

g) Sample size:

• Please explain/edit the sentence starting on line 216: “We could not communicate…”

Response: We have edited that sentence and remove the redundant part.

• Please explain/edit the sentence starting on line 218: “The sample size was adequate…” how did you come to the conclusion that the sample size was adequate?

I suggest the authors move this information to the results section.

Response: We quite concur with you. The statement is rephrased as “Assuming 20-40% of the participants might lack in improvement of knowledge and skill after the training sessions, the estimated sample size with 80% power, at a 95% confidence limit and 5% effect size/precision was 246 - 369, which was achieved for the present study evaluation”.

4. Results

h) Please note that the quality of Figures 1 and 2 is not good enough for the reader to see the numbers or legends.

Response: We regret for the inconvenience occurred, we have re-created figures of 300 dpi as per journal’s requirement for your kind consideration.

i) Line 250: please explain the meaning of the word “destruction” in this sentence.

Response: We apologies for the unclear statement, we meant that doctors suffered COVID-19 more than nurses, they also evident suffering of family members more than nurses. But we are now rephrasing the statement for a clearer message.

j) Line 268-270: please clarify this sentence.

Response: It means, in figure 2, there is an obvious improvement of knowledge score for doctors compare to nurses. We have rephrased the sentence.

k) Line 273: please change “mentioned” to “shown”.

Response: Changed

l) Table 1: please change “COVID-19 sufferer” by “COVID-19 infection”

Response: We agree with the reviewer. Corrected or changed throughout the draft.

m) Table 4:

• Please change “COVID-19 sufferer” by “COVID-19 infection”

Response: Corrected

• Please change “not-sufferer by COVID-19” by “Non-infected with COVID-19”

Response: Corrected

n) Table 5:

• Please explain the use of the term “reference” in this table.

Response: Corrected. For an independent variable, we selected a group as a base line and compared the other group(s) with the base line group to identify any association with the dependant variable by calculating the odds ratio. For each independent variable, the baseline group is the reference group (table 5).

• Please change “COVID-19 sufferer” by “COVID-19 infection”.

Response: Corrected

5. Discussion

o) Line 286-287: please clarify what you want to communicate with “…hospitals will not meet

the entire epidemic burden”.

Response: This COVID-19 pandemic has revealed the fact that health system is unprepared to tackle any health emergency and the condition is worse in developing countries.

We have rephrased the sentence and revised accordingly.

p) Line 289: please clarify to which study you refer with “the major strength is the ….”.

Response: The sentence should be “the major strength of our research paper is”. Corrected accordingly.

q) Line 308: did you also find that young people were better achievers than aged people? Please clarify this paragraph.

Response: Yes, references are included. We have further clarified in recent draft.

r) Lines 327-330: this sentence is too long and needs editing.

Response: Thanks for your suggestion, we have rephrased accordingly.

---

## [Decision Letter · Decision Letter 1]

9 Jun 2022

PONE-D-22-01969R1In-person training in COVID-19 Case management and infection prevention and control: Evaluation of healthcare professionals’ knowledge in BangladeshPLOS ONE

Dear Dr. Shahrin,

Thank you for submitting your revised manuscript to PLOS ONE. The reviewers are generally satisfied with the revisions to manuscript but some editing  concerns still remain. Specifically, please address the comments by reviewer 1 and resubmit a revised manuscript for review. 

We look forward to receiving your revised manuscript.

Kind regards,

Sanjai Kumar

Academic Editor

PLOS ONE

Journal Requirements:

Reviewers' comments:

Reviewer's Responses to Questions

**Comments to the Author**

1. If the authors have adequately addressed your comments raised in a previous round of review and you feel that this manuscript is now acceptable for publication, you may indicate that here to bypass the “Comments to the Author” section, enter your conflict of interest statement in the “Confidential to Editor” section, and submit your "Accept" recommendation.

Reviewer #1: All comments have been addressed

Reviewer #2: All comments have been addressed

2. Is the manuscript technically sound, and do the data support the conclusions?

Reviewer #1: Yes

Reviewer #2: Yes

3. Has the statistical analysis been performed appropriately and rigorously? 

Reviewer #1: Yes

Reviewer #2: Yes

4. Have the authors made all data underlying the findings in their manuscript fully available?

Reviewer #1: Yes

Reviewer #2: Yes

5. Is the manuscript presented in an intelligible fashion and written in standard English?

Reviewer #1: Yes

Reviewer #2: Yes

6. Review Comments to the Author

Reviewer #1: The manuscript “In-person training in COVID-19 Case management and infection prevention and control: Evaluation of healthcare professionals’ knowledge in Bangladesh” by Shahrin et. al., describes the importance of implementing the structured in-person training program for healthcare workers regarding case management in a pandemic.

Authors of this manuscript showed the effectiveness of in-person training program, where they compared participants’ pre and post training scores in epidemiology, clinical manifestation, management of case and infection prevention. Healthcare workers were clearly benefitted through this training program which was planned for better preparedness in controlling current or future disease outbreaks in Bangladesh.

I consider that the manuscript of Shahrin et. al., is clearly written and their claims are fully supported by well-designed study and proper statistical analysis. They obtained proper ethical approval for this study. Authors have responded to the issues found in earlier review. Authors recreated figure 1 & 2, which are now very clear to the readers. After revision with scientific editor the quality of the manuscript improved a lot.

So, I recommend the publication of this manuscript in PLOS ONE after minor correction.

But still there are some minor typographical errors, that needs to be corrected in my opinion. Like in line 143, 184, 186, 197-198, 202, 208, 226 (The approval had ID-ACT-01112).

Reviewer #2: (No Response)

7. PLOS authors have the option to publish the peer review history of their article (what does this mean?). If published, this will include your full peer review and any attached files.

Reviewer #1: **Yes: **Nirmallya Acharyya

Reviewer #2: No

---

## [Author Response · Author response to Decision Letter 1]

22 Jun 2022

Reviewer’s comments:

Reviewer #1: The manuscript “In-person training in COVID-19 Case management and infection prevention and control: Evaluation of healthcare professionals’ knowledge in Bangladesh” by Shahrin et. al., describes the importance of implementing the structured in-person training program for healthcare workers regarding case management in a pandemic.

Authors of this manuscript showed the effectiveness of in-person training program, where they compared participants’ pre and post training scores in epidemiology, clinical manifestation, management of case and infection prevention. Healthcare workers were clearly benefitted through this training program which was planned for better preparedness in controlling current or future disease outbreaks in Bangladesh.

I consider that the manuscript of Shahrin et. al., is clearly written and their claims are fully supported by well-designed study and proper statistical analysis. They obtained proper ethical approval for this study. Authors have responded to the issues found in earlier review. Authors recreated figure 1 & 2, which are now very clear to the readers. After revision with scientific editor the quality of the manuscript improved a lot.

So, I recommend the publication of this manuscript in PLOS ONE after minor correction.

Comments: Thank you very much for the encouraging comments

Minor comments: But still there are some minor typographical errors, that needs to be corrected in my opinion. Like in line 143, 184, 186, 197-198, 202, 208, 226 (The approval had ID-ACT-01112).

Comments: We have addressed the above-mentioned comments point by point and marked in underlined in the track changed version.

We hope and pray that the respected reviewer and academic editor will accept the revised drafts and allow us for the publication.

---

## [Decision Letter · Decision Letter 2]

16 Aug 2022

In-person training on COVID-19 case management and infection prevention and control:Evaluation of healthcare professionals in Bangladesh

PONE-D-22-01969R2

Dear Dr. Shahrin,

We’re pleased to inform you that your manuscript has been judged scientifically suitable for publication and will be formally accepted for publication once it meets all outstanding technical requirements.

Kind regards,

Humayun Kabir, MSc in Epidemiology

Academic Editor

PLOS ONE

Additional Editor Comments (optional):

Dear Authors,

Do some copy editing throughout the manuscript during proofreading.

For example, software STATA was written as "Stata" and "STATA". Please, choose any format.

Thanks!

Reviewers' comments:

Reviewer's Responses to Questions

**Comments to the Author**

1. If the authors have adequately addressed your comments raised in a previous round of review and you feel that this manuscript is now acceptable for publication, you may indicate that here to bypass the “Comments to the Author” section, enter your conflict of interest statement in the “Confidential to Editor” section, and submit your "Accept" recommendation.

Reviewer #1: All comments have been addressed

Reviewer #2: All comments have been addressed

2. Is the manuscript technically sound, and do the data support the conclusions?

Reviewer #1: Yes

Reviewer #2: Yes

3. Has the statistical analysis been performed appropriately and rigorously? 

Reviewer #1: Yes

Reviewer #2: Yes

4. Have the authors made all data underlying the findings in their manuscript fully available?

Reviewer #1: Yes

Reviewer #2: Yes

5. Is the manuscript presented in an intelligible fashion and written in standard English?

Reviewer #1: Yes

Reviewer #2: Yes

6. Review Comments to the Author

Reviewer #1: In this re-revised manuscript “In-person training on COVID-19 case management and infection prevention and control: Evaluation of healthcare professionals in Bangladesh” by Shahrin et. al., describes the importance of implementing the structured in-person training program for healthcare workers regarding COVID-19 case management in Bangladesh.

I consider that the manuscript of Shahrin et. al., is clearly written and their claims are fully supported by well-designed study and proper statistical analysis. They obtained proper ethical approval for this study. Authors addressed all the minor issues pointed out in the earlier review. I wish them all the very best.

So, I recommend the publication of this manuscript in the esteemed journal PLOS ONE.

Reviewer #2: (No Response)

7. PLOS authors have the option to publish the peer review history of their article (what does this mean?). If published, this will include your full peer review and any attached files.

Reviewer #1: **Yes: **Nirmallya Acharyya

Reviewer #2: No

---

## [Editor Report · Acceptance letter]

11 Sep 2022

PONE-D-22-01969R2 

In-person training on COVID-19 case management and infection prevention and control: Evaluation of healthcare professionals in Bangladesh 

Dear Dr. Shahrin:

I'm pleased to inform you that your manuscript has been deemed suitable for publication in PLOS ONE. Congratulations! Your manuscript is now with our production department. 

Kind regards, 

on behalf of

Mr. Humayun Kabir 

Academic Editor

PLOS ONE